# Survival from Cervical Cancer Diagnosed Aged 20–29 Years by Age at First Invitation to Screening in England: Population-Based Study

**DOI:** 10.3390/cancers12082079

**Published:** 2020-07-28

**Authors:** Alejandra Castanon, Daniela Tataru, Peter Sasieni

**Affiliations:** 1School of Cancer & Pharmaceutical Sciences, Innovation Hub, Guys Cancer Centre, Guys Hospital, London SE1 9RT, UK; peter.sasieni@kcl.ac.uk; 2National Cancer Registration and Analysis Service, Public Health England, London SE1 8UG, UK; daniela.tataru@phe.gov.uk

**Keywords:** cervical cancer, cervical screening, cancer intelligence, early diagnosis, survival, hazard ratios, mortality, overdiagnosis, micro-invasion, young women, screen-detected, trends

## Abstract

Age at which women are first invited to attend cervical screening in England has changed twice: in 2004, women under 25 years were no longer invited; and in 2012, first invitations were sent six months earlier (at age 24.5 years). Concomitantly, a dramatic increase in screen-detected cervical cancer was observed, and their survival had not been documented. Diagnoses of invasive cervical cancer at ages 20–29 years in 2006–2016 in England were followed until the end of 2018 for deaths. We estimated 8-year overall survival (OS) by International Federation of Gynecology and Obstetrics (FIGO) stage and age at first screening invitation. Overall and relative survival for stage IA cervical cancer for women diagnosed aged 20–29 years in England (*n* = 1905) was excellent at 99.8% (95% confidence intervals (CI): 99.4–99.9%) and 100% (95% CI: 99.7–100.1%), respectively. OS for stage IB cervical cancer (*n* = 1101) was 90.4% (95% CI: 88.3–92.2%). Survival from stage IB was worse for women diagnosed age 20–24 years compared to those diagnosed 25–29 years at diagnosis (*p* < 0.0001), but no difference was observed by age at first invitation for screening, *p* = 0.8575. OS for stage II (65.5%, 95% CI: 60.2–72.0%) and stage III+ (36.6%, 95% CI 28.4–44.7%) were poorer. Survival from stage I cervical cancer in young women in England is excellent: mortality in women with stage IA cancer is akin to that of the general population regardless of age at first invitation to screening.

## 1. Introduction

The National Health Service (NHS) Cervical Screening Program in England invites women aged 25–49 years for cytology screening every three years and women aged 50 to 64 years every five years. Over the last 15 years, the age at which women get their first invitation to attend cervical screening has changed twice: once in 2004, and then again toward the end of 2012. In 2004, the age of first cervical screening invitation in England was increased from 20 to 25.0 years [1]. In 2012, the age of sending out the first screening invitation changed once more: this time to 24.5 years (to enable women to be screened by their 25th birthday) [2]. While rates of cervical cancer in other age groups have remained stable, rates among women aged 25–29 have increased from about 10 to 22 per 100,000 between 2005 and 2015. Mortality from cervical cancer under age 30 has remained stable. Between 2013 and 2016, mortality rates per 100,000 were 0.3 at ages 20–24 years and 1.3 at aged 25–29 years [3].

Inviting women at age 25.0 years was associated with an increase of 43.7 cancers diagnosed at age 25 years per 100,000 women-years (95% confidence interval (CI): 37.4–49.9, *p* < 0.001) [4]. These extra cancers were all International Federation of Gynecology and Obstetrics (FIGO) stage IA or IB; no changes in the number of cancers diagnosed at more advanced stages (FIGO stage II+) was observed. An increase in diagnoses at age 28 among those invited at age 25 was also observed with no evidence of an increase in the stage at which these cancers were diagnosed.

Here, we aim to explore the effect of the changes to the age at first screening invitation on survival from cervical cancer diagnosed in women under the age of 30. Given the changes in age of first invitation for screening, survival was both explored by screening cohort (invited from age 20; mix of age at first invitation; invited at age 25.0 and invited at age 24.5) and by age at diagnosis (20–24.5 (or 25.0) and 25 (or 24.5)–29 years). It is worth noting that the cut-off age for the groups by age at diagnosis will depend on the age at which women were first invited to screening. Finally, we explored the survival from stage IB cervical cancer among those invited at age 24.5 to 29.9 years and grouped them based on whether the cancer was diagnosed as a result of screening or not.

## 2. Results

Between 2006 and 2016, there were 4322 eligible cervical cancers diagnosed in women aged 20–29 years. Of these cases, 20.8% either had an unknown or missing FIGO stage (*n* = 734) or a FIGO stage I not otherwise specified (*n* = 167). Forty-four percent of women were known to have a stage IA (*n* = 1905), 25% a stage IB (*n* = 1101), 6.2% a stage II (*n* = 268), and 3.4% a stage III+ (*n* = 147) cervical cancer. Overall survival at eight years for women diagnosed age 20–29 years with stage IA cervical cancer was excellent (99.8%, 95% CI: 99.4–99.9%), whereas eight-year survival for those with stage III+ cancer was just 36.6% (95% CI: 28.4–44.7%), as shown in Figure 1 and Table 1.

### 2.1. Survival from FIGO Stage IA Cervical Cancer

Among the 1905 women with stage IA cervical cancer, there were three deaths over 10,960 women-years of observation whilst lifetables suggest 2.8 were expected. Hence both observed (99.8%, 95% CI: 99.5–99.9%) and relative survival (100%, 95% CI: 99.7–100.1%) were excellent. In fact, observed survival was ≥99% regardless of age at first invitation or age at diagnosis (Table 2).

### 2.2. Survival from FIGO Stage IB Cervical Cancer

Among the 1101 women diagnosed with stage IB cervical cancer in the study, eight-year survival was 90.4% (95% CI: 88.3–92.2%, Table 3). Among those first invited to screening from age 20 years, 8-year survival was 89.9% (95% CI: 86.5–92.5%) and it was 87.3% (95% CI: 80.0–92.0%) among those invited at age 25 years (Figure 2). There was not enough follow-up to estimate 8-year survival among those invited at age 24.5 years. The difference in survival between those invited from age 20 years and those invited age 24.5 or 25 years was not statistically significant (chi2 (1) = 0.03, *p* = 0.86). However, survival was significantly worse for women diagnosed under age 24.5 (or 25.0) compared to women diagnosed 25 (or 24.5)–29 years, chi2 (1) = 18.2, *p* < 0.0001. Worst differences in 8-year survival between women diagnosed age 20–24 years compared to those diagnosed 25–29 years were observed in those first invited for screening from age 20: 75% vs. 90.7%, chi2 (1) = 5.1, *p* = 0.024 and among those first invited at age 25.0: 65.1% vs. 91.9%, chi2 (1) = 21.6, *p* < 0.0001 (Table 3).

### 2.3. Survival from FIGO Stage II or Worse Cervical Cancer

A total of 35.9% of all cervical cancers (*n* = 395/1101) diagnosed with stage IB cervical cancer were known to be screen-detected cancers, 29.5% (*n* = 325) had no information on screening status, and 34.6% (*n* = 381) were not screen detected. Here, we restricted the analysis to include only those cases that would have been due a screening test at the time of diagnosis, which we classified in three groups. Hence only 70% (*n* = 278/395) of women identified as being screen-detected and 31% (*n* = 120/381) of those not screen-detected were included in this analysis. Eight-year survival was similar among all screen-detected groups. It was 94.6% (95% CI: 88.9–97.4%) in the group of women first invited at ages 24.5–25.0 years and diagnosed age of 24.5–25.9. It was 94.5% (95% CI: 88.1–97.5%) in women first invited at ages 24.5–25.0 years and diagnosed age 27.5–28.9 (around their second invitation). Among those diagnosed age 24.5–29.9 years during the Jade Goody period, survival was 94.1% (95% CI: 78.5–98.5%). The much-publicized diagnosis and death from cervical cancer of celebrity Jade Goody led to a 70% increase in attendance to screening (between September 2008 and June 2009) in England, leading to a substantial increase in the diagnosis of cervical cancer [5]. Eight-year survival among women who were not screen-detected was 88% (95% CI: 80–93%). Survival among those following screen-detected cancer was borderline significantly different from survival among women whose cancers were not screen-detected, chi2 (1) = 4.4 *p* = 0.04.

Survival among the 268 women with FIGO stage II cervical cancer was similar at 5-years (67.7%, 95% CI: 61.5–73.1%) and eight years post diagnosis (66.5%, 95% CI: 60.2–72.0%, Table 4). There was no difference in eight-year survival from stage II cervical cancer by age at first invitation to screening: 69.7% (95% CI: 59.5–77.9%) in those invited from age 20 and 67.5% (95% CI: 58.6–74.8%) in those invited at age 24.5 or 25.0, *p* = 0.6877. Survival among the 147 women with stage III or worse cervical cancer was poor (36.6%, 95% CI: 28.4–44.7%). There was no evidence that survival from stage II or worse cervical cancer differed by age at first invitation, regardless of age at diagnosis, *p* = 0.7083.

## 3. Discussion

Five-year survival from stage IA cervical cancer in women under age 30 in England diagnosed during 2006–2016 was excellent (≥99%), regardless of the age at first invitation for screening. There is no evidence that mortality rates within eight years of stage IA cervical cancer diagnosis was any worse than among all women aged 20–29 years living in England. Although 8-year survival from stage IB cervical cancer in women aged 20 to 29 was 90%, survival was significantly poorer for those diagnosed under age 25 (between 65–75%) compared to those diagnosed at ages 25–29 (around 92%). There was also evidence that survival from IB cervical cancer was better among the screen-detected women. Survival for stage II or worse cervical cancer was poorer. However, since most of these women will be unscreened, one would not expect differences in survival by age of invitation and indeed, there is no difference.

Strengths of the study include the use of population-based data from the National Cancer Registration Dataset, which ensures the inclusion of almost all cervical cancers diagnosed in England over the period of study.

Although cancer registry data are not linked to screening history information, there were data on whether the cancer was screen-detected or not, which enabled us to carry out a sub analysis. The results presented here suggest better survival from FIGO IB cervical cancer when it is screen-detected. A difference in survival between screen-detected and symptomatic cancers of all stages has been reported previously [6]. Survival of screen-detected cancers is subject to lead-time bias. Nevertheless, the fact that 8-year survival for stage IB cervical cancer was more than 95% is excellent news for patients and has not, as far as we are aware, been reported before. Survival among the screen-detected stage IB cancer was not substantially greater than survival by age at invitation. It can be argued that stage IB cancers should be homogeneous in their survival since without lead time, they might have been diagnosed at stage IIA. This appears to be true. We did not explore survival by screen-detected status for stage IA or stage II+ cancers because there were only three deaths in 1905 women with stage IA cancer and there were so few screen-detected stage II or worse cervical cancers.

The proportion of women whose cancer stage was FIGO I or unknown differed between screening cohorts and could be a source of bias. The proportion of cancers with FIGO I or unknown stage ranged from 33% among women invited from age 20 years to 9% among those invited at age 24.5 years. The lack of staging information is directly linked to the stage at diagnosis: the more advanced cancers are the least likely to be staged [7]. In this study, 38% of cancers diagnosed in women under age 24.5 years were stage II or worse compared to 10% of those diagnosed in women aged 24.5–29 years.

Statistics for England report 5-year net survival from cervical cancer diagnosed between 2013 and 2017 at ages 15–44 of 97.2% (95% CI: 96.6–97.8%) for stage I cervical cancer and of 72.0% (95% CI: 67.2–76.8%) for stage II and of 89.1% (95% CI: 88.2–90.2%) overall [8]. The Surveillance, Epidemiology, and End Results (SEER) Program statistics reported by the American Cancer Society [9] show 5-year relative survival rates for all ages (2008–2014) of 92% for localized (stage I) and 56% for regional (stage II, III, and IV) cervical cancer. An older publication [10] using SEER data from 1988 to 2001 reports 5-year survival by age and FIGO stage. Survival from stage IA cervical cancer at age 20–49 was 98.3%, 89.4% for stage IB, 61.2% for stage II, 50.9% for stage III, and 20.9% for stage IV. This is the first time that survival in England for cervical cancer stage IA is reported separately from those with stage IB. Furthermore, we focused on women diagnosed aged 20–29 years.

The published literature supports the view that younger age (usually those under age 50) and early stage (i.e., localized) at diagnosis are prognostic factors for improved survival from cervical cancer and that improved survival over time can be attributed to the effect of screening on both prognostic factors [11,12,13]. Here, poorer survival was observed for stage IB cervical cancer in women under age 25, regardless of the age at first invitation.

It appears that the change in policy on the age at which women are first invited to screening has increased diagnoses of cervical cancers by 45% [4] without significantly increasing the number of deaths. We note that only 3.3% (*n* = 113) of women eligible for analysis were also eligible for human papilloma virus vaccination, so vaccination cannot have had a substantial impact on incidence in this cohort. Localized cervical cancer is treated using fertility preserving methods such as cone biopsy or LLETZ (large loop excision of the cervical transformation) [14]. LLETZ is also the most common treatment for a diagnosis of pre-cancerous lesions.

## 4. Materials and Methods

Data on all cervical cancer tumors (ICD10 code C53) diagnosed between January 2006 and December 2016 in women aged 20 to 29 years and resident in England were extracted from the cancer registration dataset produced by the National Cancer Registration and Analysis Service (NCRAS) [15]. Date of diagnosis, age, and FIGO stage at diagnosis, cervical cancer screening information, date of death (where applicable), and vital status on 31 December 2018 were extracted.

Twenty-six women had two cervical cancer tumors diagnosed during the study period. Only the first diagnosis was included, and when both tumors were diagnosed on the same day, the tumor with non-missing or more advanced stage was retained. One death certificate only (DCO) and seven women with data quality issues were excluded from the analysis.

Data were available on whether a cervical cancer was screen detected or not, but we did not have access to full screening history data, and hence did not know the exact dates from which women would have been invited from age 25. To explore the effect of age at first invitation to screening on survival women diagnosed with cervical cancer were grouped in one of the following four screening cohort groups:Women born on or before 25 August 1984 (Invited from age 20)Women born 26 August 1984 to 3 November 1985 (Mix of age at first invitation)Women born 4 November 1985 to 12 June 1988 (Invited at age 25.0)Born on or after 12 June 1988 (Invited at age 24.5)

To explore survival by age at diagnosis, we grouped stage IA cervical cancers in four age groups (20–24.4, 24.5–26, 27–28, and 29) so that we could compare survival to the general population using lifetables. For stage IB or worse cervical cancer, age at diagnosis was grouped as follows:For women invited at age 24.5 years: 20.0–24.49 years and 24.5–29.9 years.For all other women: 20.0–24.9 years and 25.0–29.9 years.

This is reported in the results as 20–24.5 (or 25) and 25 (or 24.5)–29.

To explore survival among those diagnosed with stage IB cervical cancer in more detail, we focused on women invited at age 24.5 to 29.9 years and grouped them based on whether the cancer was diagnosed as a result of screening or not. We used the information in the National Cancer Registration Dataset on whether or not a cancer was screen-detected. We divided these women with screen-detected cervical cancer into three groups as follows:Women with age at diagnosis 24.5–25.9 years, which were recorded in the cancer registry as being screen-detected or born on or after 4 November 1985 (i.e., with first invitation at age 24.5 or 25.0 years). Women diagnosed between 1 September 2008 to 30 June 2009 were excluded (because of the increase in women attending screening as a result of the much-publicized diagnosis and death from cervical cancer of celebrity Jade Goody in March 2009, which led to a substantial increase in the diagnosis of cervical cancer [5]).Women with age at diagnosis of 27.5–28.9 years, which were recorded in the cancer registry as being screen-detected or born on or after 4 November 1985 (i.e., with first invitation at age 24.5 or 25.0 years). Women diagnosed during the Jade Goody period were excluded.Women with age at diagnosis of 24.5–29.99 and diagnosed during the Jade Goody period: 1 September 2008 to 30 June 2009.Not screen-detected included women in the three categories above who were known not to have been screen-detected.

The main outcome was 8-year overall survival among women diagnosed with stage IA or stage IB cervical cancer by age at first invitation for screening. Secondary analyses explored 8-year survival among those diagnosed with stage II or worse cervical cancer and whether being screen-detected influenced survival from stage IB cervical cancer.

All-cause mortality was used to calculate survival at 1, 5, and 8 years by FIGO stage, age at diagnosis, and age of first invitation to screening. The censoring date for deaths was the 31 December 2018 or date of death, if earlier. Survival, 95% confidence intervals, women-years at risk, and number of deaths were calculated using Kaplan–Meier survival estimate (using Stata sts command, Stata 15.1 Copyright 1985–2017 StataCorp LLC, StataCorp 4905 Lakeway Drive, College Station, TX, USA). When there were no deaths, we used a one-sided 97.5% confidence interval based on a poison observation of 0 deaths in int (PY/K) people, where PY is the number of person-years of follow-up up to K years. Log-rank test was used to assess differences in survival.

National life tables for England based on data for the years 2015–2017 [16] were used to estimate relative survival from stage IA cervical cancer. We achieved this by dividing the observed survival by the expected survival, and similarly the upper and lower confidence intervals of the observed survival by the expected survival.

## 5. Conclusions

Cervical cancer screening services across the globe have been severely disrupted during the COVID-19 pandemic. The resulting delays to routine screening are likely to result in an increased number of diagnoses of screen-detected (and maybe even symptomatic) cancer. Given the excellent eight-year survival in women diagnosed with stage IA cancer under the age of 30 (regardless of the age of first invitation to screening), that the majority of stage IA cervical cancer is asymptomatic, and that treatment can preserve fertility, diagnoses of stage IA cervical cancer in women under age 30 should be considered a success of the screening program.

## Figures and Tables

**Figure 1 cancers-12-02079-f001:**
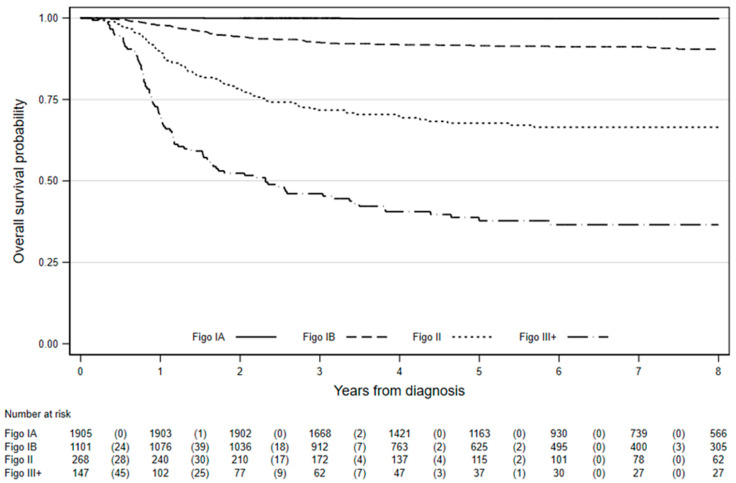
Cervical cancer survival estimates in women aged 20–29 in England by International Federation of Gynecology and Obstetrics (FIGO) stage. Numbers in parenthesis are deaths in the interval.

**Figure 2 cancers-12-02079-f002:**
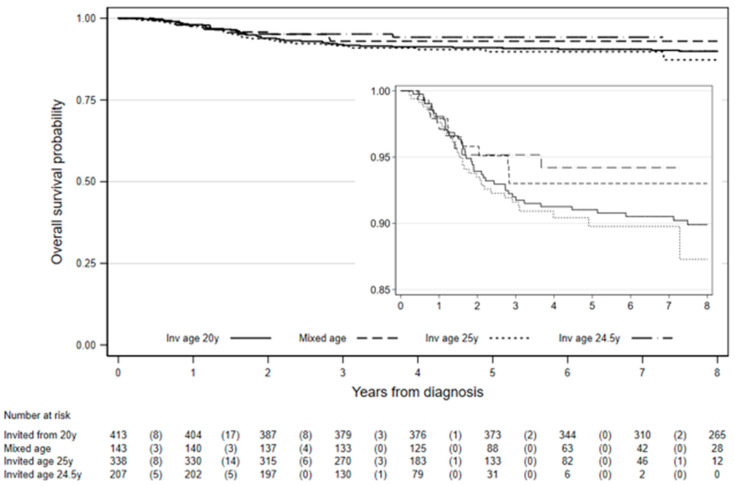
Cervical cancer survival estimates from FIGO stage IB by age at first invitation to screening in England. Numbers in parenthesis are deaths in the interval.

**Table 1 cancers-12-02079-t001:** Eight-year cervical cancer survival by International Federation of Gynecology and Obstetrics (FIGO) stage in women aged 20–29 in England.

FIGO Stage	Number of Cases (*n*)	Proportion	Eight-Year Survival (95% CI)
IA	1905	44.0%	99.8 (99.4 to 99.9)
IB	1101	25.5%	90.4 (88.3 to 92.2)
I NOS *	167	3.9%	95.8 (91.4 to 98.0)
NK *	734	17.0%	85.8 (83.0 to 88.2)
II	268	6.2%	66.5 (60.2 to 72.0)
III+	147	3.4%	36.6 (28.4 to 44.7)

* Not otherwise specified (NOS), Not Known (NK).

**Table 2 cancers-12-02079-t002:** One-, five- and eight- year cervical cancer survival estimates (95% CI) for Stage IA by screening cohort and age at diagnosis.

Age Ranges	Number of Cases (*n*)	1-Year Survival	5-Year Survival	8-Year Survival
All Ages	1905	100 (99.8–100)	99.8 (99.5–99.9)	99.8 (99.5–99.9)
**Survival by Screening Cohort**				
Invited from Age 20 years	637	100 (99.4–100)	99.8 (98.9–100)	99.8 (98.9–100)
Mixed Age at Invitation	259	100 (98.6–100)	99.6 (97.3–100)	99.6 (97.3–100)
Invited from Age 25.0 years	576	100 (99.4–100)	100 (99.3–100)	100 (99.0–100)
Invited from Age 24.5 years	433	100 (99.1–100)	99.8 (98.4–100)	99.8 (98.4–100)
**Survival by Age at Diagnosis**				
20–24.4 years	62	100 (94.1–100)	100 (93.9–100)	100 (93.4–100)
24.5–26 years	947	100 (99.6–100)	99.8 (99.1–99.9)	99.8 (99.1–99.9)
27–28.9 years	640	100 (99.4–100)	100 (99.3–100)	100 (99.2–100)
29 years	256	100 (98.6–100)	99.5 (96.6–99.9)	99.5 (96.6–99.9)

**Table 3 cancers-12-02079-t003:** One-, five-, and eight-year Kaplan–Meier survival estimates (95% CI) for Stage IB by screening cohort and age at diagnosis.

Age Ranges	Number of Cases (*n*)	1-Year Survival	5-Year Survival	8-Year Survival
All ages	1101	97.8 (96.8–98.5)	91.5 (89.6–93.0)	90.4 (88.3–92.2)
**Survival by Screening Cohort**				
Invited from Age 20 years				
All Ages at Diagnosis	413	98.1 (96.2–99.0)	91.0 (87.8–93.4)	89.9 (86.5–92.5)
20–24 years	20	95.0 (69.5–99.3)	75.0 (50.0–88.8)	75.0 (50.0–88.8)
25–29 years	393	98.2 (96.3–99.1)	91.8 (88.7–94.2)	90.7 (87.3–93.2)
Mix of Ages at First Invitation				
All ages at Diagnosis	143	97.9 (93.6–99.3)	93.0 (87.4–96.2)	93.0 (87.4–96.2)
20–24 years	11	100 (66.5–100)	100 (66.5–100)	100 (66.5–100)
25–29 years	132	97.7 (93.1–99.3)	92.4 (86.4–95.9)	92.4 (86.4–95.9)
Invited at Age 25.0 years				
All Ages at Diagnosis	338	97.6 (95.3–98.8)	89.8 (85.7–92.7)	87.3 (80.0–92.0)
20–24 years	32	87.5 (70.0–95.1)	68.8 (49.7–81.8)	65.1 (45.9–79.0)
25–29 years	306	98.7 (96.6–99.5)	91.9 (87.8–94.7)	91.9 (87.8–94.7)
Invited at Age 24.5 years				
All Ages at Diagnosis	207	97.6 (94.3–99.0)	94.2 (89.6–96.8)	N/A
20–24.4 years	34	94.1 (78.5–98.5)	91.2 (75.1–97.1)	N/A
24.5–29 years	173	98.3 (94.7–99.4)	94.7 (89.3–97.4)	N/A
**Survival among Women Known to Be Screen-Detected**				
Diagnosed 24.5–25.9 years Invited 24.5–25.0 years	132	99.2 (94.7–99.9)	94.6 (88.9–97.4)	94.6 (88.9–97.4)
Diagnosed 27.5–28.9 years Invited 24.5–25.0 years	112	99.1 (93.8–99.9)	94.5 (88.1–97.5)	94.5 (88.1–97.5)
Diagnosed 24.5–29.9 years Invited from 20 years during the Jade Goody Period	34	100 (89.2–100)	97.1 (80.9–99.6)	94.1 (78.5–98.5)
Not Screen-Detected	120	96.7 (91.4–98.7)	88.1 (80.0–93.1)	88.1 (80.0–93.1)

**Table 4 cancers-12-02079-t004:** One-, five-, and eight-year Kaplan–Meier survival estimates (95% CI) for Stage II+ by screening cohort.

Age Ranges	Number of Cases (*n*)	1-Year Survival	5-Year Survival	8-Year Survival
Stage II				
All Ages	268	89.6 (85.2–92.7)	67.7 (61.5–73.1)	66.5 (60.2–72.0)
Invited from 20 y	96	91.7 (84.0–5.7)	70.8 (60.6–78.8)	69.7 (59.5–77.9)
Mix Age at First Invitation	29	86.2 (67.3–94.6)	60.9 (40.3–76.2)	55.3 (34.1–72.2)
Invited at Age 24.5 or 25.0 years	143	88.8 (82.4–93.0)	67.5 (58.6–74.8)	67.5 (58.6–74.8)
Stage III+				
All Ages	147	69.4 (61.3–76.2)	37.8 (29.7–45.8)	36.6 (28.4–44.7)
Invited from 20 years	53	69.8 (55.5–80.3)	39.6 (26.6–52.4)	37.7 (24.9–50.5)
Mix Age at First Invitation	30	60.0 (40.5–75.0)	25.7 (11.6–42.4)	25.7 (11.6–42.4)
Invited at Age 24.5 or 25.0 years	64	73.4 (60.8–82.6)	43.0 (30.0–55.3)	43.0 (30.0–55.3)

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
