# Peer review of "Survival from Cervical Cancer Diagnosed Aged 20–29 Years by Age at First Invitation to Screening in England: Population-Based Study"

_cancers, 2020, doi:10.3390/cancers12082079_

Round 1
Reviewer 1 Report
I will say that I still don't like journals that show the results before the methods - but that is not the fault of the authors.
However, it does mean that the Grouping and categorisation of Young/very young etc. should be spelled out a little more clearly in the Results section again. It means also the motivation for some of the groupings were unclear until reading the later sections also. There are quite a few results and analyses - perhaps some subheadings would be useful.
I have some minor comments for improvements:
- I think the motivation for looking at this subgroup and the groupings could be strengthened in the introduction. Why is a focus on this younger age-group of great interest - a bit more context.
- The authors mention the difficulty of lead time on the comparison of survival estimates because of screening. How reliable is the coding of the screen detected variable in the registry data? - has any study been done to understand if this is misclassified at all?
- The final line of the conclusion: survival isn't the only metric of concern of course, some caution here.
Very minor comments (up to the authors disgression): The default Stata graph scheme is not the prettiest. Possible suggestions for quick improvements: a) change the ylabel angle for better readability, b) change the graphregion to white, c) change from the default s2color scheme, d) move the legend into the plotregion to allow more space for the graph, e) add a ytitle to the graphs.
Author Response
We thank the reviewer for his comments which have helped clarify the manuscript. Our detailed response is below.
- -I will say that I still don't like journals that show the results before the methods - but that is not the fault of the authors. However, it does mean that the Grouping and categorisation of Young/very young etc. should be spelled out a little more clearly in the Results section again. It means also the motivation for some of the groupings were unclear until reading the later sections also.
Response: We apologize for not accounting for the fact that the methods come after the results in the journal template. We appreciate that the manuscript must have been difficult to read. We have endeavoured to clarify groupings and methods at the end of the introduction and in the results.
We have removed the categorisation Young/very young and instead used the ages in each group. Although this may be harder to read, we hope the reviewer agrees it is ultimately clearer to the reader.
- There are quite a few results and analyses - perhaps some subheadings would be useful.
Response: We have added 3 sub-headings to the results
- I think the motivation for looking at this subgroup and the groupings could be strengthened in the introduction. Why is a focus on this younger age-group of great interest - a bit more context.
Response: To clarify the motivation for the various groupings we have added the following paragraph to the introduction on the groupings and the reasons why age ranges vary depending on the age at which women were first invited.
“Here we aim to explore the effect of the changes to the age at first screening invitation on survival from cervical cancer diagnosed in women under the age of 30. Given the changes in age of first invitation for screening survival was both explored by screening cohort (invited from age 20; mix of age at first invitation; invited at age 25.0 and invited at age 24.5) and by age at diagnosis (20-24.5(or 25.0) and 25(or 24.5)-29y). It is worth noting that the cut-off age for the groups by age at diagnosis will depend on the age at which women were first invited to screening. Finally we explore survival from stage IB cervical cancer among those invited at age 24.5 to 29.9y and grouped them based on whether the cancer was diagnosed as result of screening or not.”
Further the methods were clarified as follows:
For stage IB or worse cervical cancer age at diagnosis was grouped as follows:
- For women invited at age 24.5y: 0-24.49y and 24.5-29.9y,
- For all other women: 0-24.9y and 25.0-29.9y.
This is reported in the results as 20-24.5(or 25) and 25(or 24.5)-29.
- The authors mention the difficulty of lead time on the comparison of survival estimates because of screening. How reliable is the coding of the screen detected variable in the registry data? - has any study been done to understand if this is misclassified at all?
Response: Although screen-detected status is missing for many women, the screening status information held by NCRAS was obtained directly from the Cancer Screening Program through a data exchange process. The screening programme defines screen detected as ‘Detected after a diagnostic process that began with a cytology test taken up to three months before the test due date or up to six months after the test was due’. To increase the accuracy, in our manuscript we chose to use a narrower definition of the screen detected category by further limiting the screen-detected group to those women that would have been due a screening test at the time of diagnosis.
In a previous version of the manuscript we intended to make our own definition of screen-detected cancer by having a composite category which included all those identified as screen-detected in the cancer registry dataset and also include anyone diagnosed at ages at which a woman is likely to have been invited for screening. Using this categorisation provided similar results to those shown in the manuscript. It was felt that this indicated the robustness of the cancer registry data and that the addition of the additional definitions did not add value.
- The final line of the conclusion: survival isn't the only metric of concern of course, some caution here.
Response: We have reworded the final paragraph as follows
‘Given the excellent eight-year survival in women diagnosed with stage IA cancer under the age of 30 (regardless of the age of first invitation to screening), that the majority of stage IA cervical cancer is asymptomatic and that treatment can preserve fertility, diagnoses of stage IA cervical cancer in women under age 30 should be considered a success of the screening programme.’
- Very minor comments (up to the authors digression): The default Stata graph scheme is not the prettiest. Possible suggestions for quick improvements: a) change the ylabel angle for better readability, b) change the graphregion to white, c) change from the default s2color scheme, d) move the legend into the plotregion to allow more space for the graph, e) add a ytitle to the graphs.
Response: We have updated the graphs as per the reviewers suggestions.
Reviewer 2 Report
I would like to thank the authors for the opportunity to review their manuscript. This report examines whether the age of first screening invitation is associated with the prognosis of cervical cancer. First of all, foreigners are not familiar with the transition of screening policy for cervical cancer in the UK. The classification of groups A, B, C is complicated for us and difficult to understand. It is necessary to make materials that help understanding, such as supplemental figures. Similarly, Jade-Goody period is an unfamiliar and needs explanation. In addition to these problems, I have some questions.
- The conclusions of this articlet should be clearly stated. At first I thought the conclusion was that the first age of invitation was not associated prognosis of cervical cancer, but the authors emphasize that the prognosis is very good in stage IA. I don't realize which is the main conclusion. In addition, the impact of COVID-19 is discussed in conclusion part. It's unclear what authors want to appeal most in this article.
- Not only changes in the number of cases of cervical cancer but also changes in the screening rate among English should be described. Foreigners do not know how much the rate of cervical screening increased during the Jade-Goody period. I think it is very important information.
- (L87) Authors described that there is a significant difference in prognosis between the Screening detected group and the not screening group, but they should described whether it is good or bad. Probably the screening detected group has a good prognosis, but this fact should be clearly described.
- (L135) This report is described to be the first report about prognosis of stage IA cervical cancer in the UK. However, there are many similar reports before, so it might not be an important finding.
Author Response
We thanks the reviewer for their comments, they have helped improve the clarity of the manuscript. Our detailed response to the points raised is below.
- I would like to thank the authors for the opportunity to review their manuscript. This report examines whether the age of first screening invitation is associated with the prognosis of cervical cancer. First of all, foreigners are not familiar with the transition of screening policy for cervical cancer in the UK. The classification of groups A, B, C is complicated for us and difficult to understand. It is necessary to make materials that help understanding, such as supplemental figures. Similarly, Jade-Goody period is an unfamiliar and needs explanation. In addition to these problems, I have some questions.
Response: We thank the reviewer for pointing out that we have not been clear on our motivation for the groupings and on the labelling of these groups. To clarify this we have
1) added a paragraph to the introduction (see response to reviewer 1);
2) have removed the use of groups A, B , C from the results and table 3. We also added the following text:
“We further classify these 395 screen-detected cancers in three groups with the aim to identify those that would have been due a screening test at the time of diagnosis. Hence only 70%(n=278/395) of women identified as being screen-detected and 31%(n=117/381) of those not screen-detected were included this analysis.”
To clarify the situation with Jade Goody we added to the results:
“Among those diagnosed age 24.5-29.9y during the Jade Goody period- survival was 94.1% (95%CI: 78.5% to 98.5%). The much-publicised diagnosis and death from cervical cancer of celebrity Jade Goody led to a 70% increase in attendance to screening in England leading to a substantial increase in diagnosis of cervical cancer (between September 2008 and June 2009).
- The conclusions of this article should be clearly stated. At first I thought the conclusion was that the first age of invitation was not associated prognosis of cervical cancer, but the authors emphasize that the prognosis is very good in stage IA. I don't realize which is the main conclusion. In addition, the impact of COVID-19 is discussed in conclusion part. It's unclear what authors want to appeal most in this article.
Response: Our main conclusion is that survival from stage IA cervical cancer is excellent regardless of the age at which screening is first offered or the age at which the cancer is diagnosed. The text in the abstract has been modified to reflect this better.
There is no doubt that the one of the impacts of COVID on cervical cancer diagnosis through delays to screening will be an increase in stage IA cervical cancer that could have been treated as CIN3. It is important to show that survival from this cancer does not differ from that of the general population in this age group. The results from this study should reassure clinicians and policy makers that possible increases in diagnoses of stage IA cancer can still be considered successes even if the cancer was not prevented (which is the aim of the screening programme).
- Not only changes in the number of cases of cervical cancer but also changes in the screening rate among English should be described. Foreigners do not know how much the rate of cervical screening increased during the Jade-Goody period. I think it is very important information.
Response: We had added the following paragraph to highlight the impact on attendance that Jade Goody had
“The much-publicised diagnosis and death from cervical cancer of celebrity Jade Goody led to a 70% increase in attendance to screening in England leading to a substantial increase in diagnosis of cervical cancer (between September 2008 and June 2009).”
- (L87) Authors described that there is a significant difference in prognosis between the Screening detected group and the not screening group, but they should described whether it is good or bad. Probably the screening detected group has a good prognosis, but this fact should be clearly described.
Response: We thanks the reviewer for pointing this out. We have now made the results clearer by adding the survival among the non screen-detected group to the results and Table 3.
“Eight-year survival among women who were not screen-detected was 88% (95% CI: 80% to 93%). Survival among following screen-detected cancer was borderline significantly different from survival among women who were not screen-detected, chi2(1) 4.4 p=0.04.”
In the discussion we have added
“Results presented here suggest better survival when FIGO IB cervical cancer is screen-detected. A difference in survival between screen-detected and symptomatic cancers of all stages has been reported previously.6”
- (L135) This report is described to be the first report about prognosis of stage IA cervical cancer in the UK. However, there are many similar reports before, so it might not be an important finding.
Response: We know of no other reports that quantify survival from stage IA separately from survival of stage IB cervical cancer in this age group. Most report on the combined (stage I) survival or report survival for stage IA and stage IB at all ages. National statistics report wide age ranges including women in their 40s.
Furthermore we have not seen any reports by age of first invitation nor by screen-detected status.
Reviewer 3 Report
This is a small, well designed paper on survival from cervical cancer in a young group of women 20-29 years of age. It is promoting systematic screening and early detection as main reasons for success. The language is good, tables are informative. I recommend this paper for publication.
Author Response
- This is a small, well designed paper on survival from cervical cancer in a young group of women 20-29 years of age. It is promoting systematic screening and early detection as main reasons for success. The language is good, tables are informative. I recommend this paper for publication.
Response: We thank the reviewer for the kind comments.
Round 2
Reviewer 2 Report
I read the Revised version interestingly. I think you made the appropriate fix. The article is easy to understand even for foreigners. The issues have been sorted out and made easier to understand. Thank you very much for your appropriate response.